# Presence of known feline *ALMS1* and *MYBPC3* variants in a diverse cohort of cats with hypertrophic cardiomyopathy in Japan

Noriyoshi Akiyama[1,2], Ryohei Suzuki[3]*, Takahiro Saito[3], Yunosuke Yuchi[3], Hisashi Ukawa[1,2], Yuki Matsumoto[1,2,4]*

1 Research and Developmental Division, Anicom Insurance Inc., Yokohama, Kanagawa, Japan, 2 Genetic Testing Section, Anicom Pafe Inc., Yokohama, Kanagawa, Japan, 3 Laboratory of Veterinary Internal Medicine, Department of Veterinary Clinical Medicine, School of Veterinary Medicine, Faculty of Veterinary Science, Nippon Veterinary and Life Science University, Musashino, Tokyo, Japan, 4 Data Science center, Azabu University, Sagamihara, Kanagawa, Japan

* ymatsumoto.ac@gmail.com (YM); ryoheisuzuki@nvlu.ac.jp (RS)

**Data Availability Statement:** All relevant data are within the paper and its Supporting Information files.

## Abstract

Hypertrophic cardiomyopathy (HCM) is the most common heart disease in cats with a suspected genetic origin. Previous studies have identified five HCM-associated variants in three genes (Myosin binding protein C3: *MYBPC3* p.A31P, p.A74T, p.R820W; Myosin heavy chain 7: *MYH7* p.E1883K; Alstrom syndrome protein 1: *ALMS1* p.G3376R). These variants are considered breed-specific, with the exception of *MYBPC3* p.A74T, and have rarely been found in other breeds. However, genetic studies on HCM-associated variants across breeds are still insufficient because of population and breed bias caused by differences in genetic background. This study investigates the ubiquitous occurrence of HCM-associated genetic variants among cat breeds, using 57 HCM-affected, 19 HCM-unaffected, and 227 non-examined cats from the Japanese population. Genotyping of the five variants revealed the presence of *MYBPC3* p.A31P and *ALMS1* p.G3376R in two (Munchkin and Scottish Fold) and five non-specific breeds (American Shorthair, Exotic Shorthair, Minuet, Munchkin and Scottish Fold), respectively, in which the variants had not been identified previously. In addition, our results indicate that the *ALMS1* variants identified in the Sphynx breed might not be Sphynx-specific. Overall, our results suggest that these two specific variants may still be found in other cat breeds and should be examined in detail in a population-driven manner. Furthermore, applying genetic testing to Munchkin and Scottish Fold, the breeds with both *MYBPC3* and *ALMS1* variants, will help prevent the development of new HCM-affected cat colonies.

## Introduction

Hypertrophic cardiomyopathy (HCM) is the most common type of heart disease in cats, with an estimated prevalence of approximately 15% in the asymptomatic cat population [1–5]. Most HCM-affected individuals do not appear symptomatic immediately. However, HCM in

**Funding:** This study was supported by Research Foundation of Anicom Insurance Inc. (Japan), Anicom Specialty Medical Institute Inc. (Japan), Anicom Pafe Inc. (Japan), Nippon Veterinary and Life Science University (Japan) and Azabu University (Japan). The funders had no role in study design, data collection and analysis, decision to publish, or preparation of the manuscript. Noriyoshi Akiyama and Yuki Matsumoto received a salary from Anicom Specialty Medical Institute Inc. and Anicom Pafe Inc. Yuki Matsumoto also received a salary from Azabu University. Ryohei Suzuki received a salary from Nippon Veterinary and Life Science University. Hisashi Ukawa received a salary from Anicom Pafe Inc. The specific roles of these authors are articulated in the 'author contributions' section.

**Competing interests:** UH is employee of Anicom Pafe Inc., a DNA testing company which will offer commercial testing for the variant described in this study. NA and YM are employees of Anicom Insurance Inc. and Anicom Specialty Medical Institute Inc., both are sister companies of Anicom Pafe Inc. This does not alter our adherence to PLOS ONE policies on sharing data and materials. There are no patents, products in development or marketed products associated with this research to declare.

cats often causes congestive heart failure and arterial thromboembolism, and a minority of cats die suddenly without clinical signs [6–8]. Feline HCM is diagnosed not only by echocardiography, which demonstrates several regional hypertrophic patterns (predominantly including the left ventricle), but also by the absence of diseases that cause cardiomyopathy phenotypes, such as hypertension and hyperthyroidism [9]. Establishing an HCM diagnosis can sometimes be challenging because of the necessary differentiation between various phenotypic categories.

Most cats with HCM are randomly bred; otherwise, HCM is often associated with some specific breeds, including Maine Coon, Ragdoll, American Shorthair, British Shorthair, Persian, Bengal, Sphynx, Norwegian Forest cat, and Scottish Fold [10–15]. The familial incidence of this disease has also been confirmed, suggesting a genetic etiology [16]. However, only a few HCM-related variants have been identified in cats.

Three genetic variants have indicated the association with feline HCM in one of the sarcomeric genes, the Myosin binding protein C3 (*MYBPC3*) gene. *MYBPC3* p.A31P was the first identified HCM-associated variant in a familial HCM colony of Maine Coon [17]. The second variant, *MYBPC3* p.R820W, was shown to be associated with HCM in Ragdoll cats [10] These two mutations are widely recognized as HCM-causative mutations, and homozygotes of each mutation have a higher risk of developing HCM [18]. Another study proposed HCM association with the *MYBPC3* p.A74T variant; however, this mutation was considered unrelated to cardiomyopathy in recent follow-up studies [19, 20]. Next, Myosin heavy chain 7 (*MYH7*) p. E1883K variant was detected in an HCM-affected Domestic Shorthair cat [21]. This amino acid mutation has also been reported to be associated with the human HCM [22]. Finally, a mutation recently identified in the HCM-affected Sphynx population belongs to Alstrom syndrome protein 1 (*ALMS1*) p.G3376R and is considered Sphynx-specific because it is seldom found in more than 200 non-Sphynx cats [23]. Characteristically, *ALMS1* is a ubiquitously expressed non-sarcomeric gene [24]. Therefore, studies on HCM-associated genetic variants can help understand HCM risk in advance using genetic testing. However, these mutations are determined to be breed-specific, and the feline HCM guidelines of the Consensus Statements of the American College of Veterinary Internal Medicine do not recommend genetic testing for *MYBPC3* p.A31P and p.R820W in cats not belonging to Maine Coon and Ragdoll breed [9].

HCM-associated variants have been analyzed occasionally in non-specific breeds that often lead to HCM development. Furthermore, studies in the U.S. have yet to find evidence of variants being universally present among breeds [19, 23, 25]. Additionally, only a single study of a large European feline population reported the presence of *MYBPC3* p.A31P in one of the two British Longhair cats besides Maine Coon; however, the findings appeared insufficient to contradict the conclusion that the mutation is specific to Maine Coon [26]. Moreover, the information available on HCM-related variants in a wide variety of breeds is still inadequate because of population and breed biases. For example, genome-wide comparisons of 13 pedigrees and random-bred populations between the U.S. and Japan showed differences in the genetic structure of the majority of pedigrees [27]. Therefore, the genetic structure of breeds often differs between populations owing to differences in the historical lineage and inbreeding levels. In this study, we aimed to verify the spread of HCM-associated variants across broad cat breeds. In addition, we addressed the issue of population bias by using a Japanese population instead of a Western population, which is commonly used in similar studies.

## Materials and methods

### Ethics statement

All clinical examinations were performed in accordance with the Guidelines for Institutional Laboratory Animal Care and Use of Nippon Veterinary and Life Science University in Japan

(No. R2-4). Written informed consent authorizing participation in the study was obtained from the cat owners. All swab samples from cats were taken with their consent for genetic testing and were approved by the Ethics Committee of Anicom Speciality Medical Institute Inc. (No. 2022–06 & No. 2022–02).

## Phenotyping of selected animals

Seventy-six client-owned cats (63 purebred cats belonging to 13 breeds and 13 Japanese random-bred) underwent complete physical examination, electrocardiography, thoracic radiography, blood pressure measurement, and transthoracic echocardiography. Among them, 57 cats (52 purebred cats belonging to 13 breeds and 5 Japanese random-bred) were diagnosed with HCM (HCM-affected group), while the other 19 cats (11 purebred cats belonging to 4 breeds and 8 Japanese random-bred) were not affected by HCM (non-HCM group). We diagnosed HCM based on echocardiographic evidence of left ventricular (LV) hypertrophy and the absence of other diseases that cause LV hypertrophy. Echocardiographic LV hypertrophy was confirmed according to the LV wall thickness of $\geq 6$ mm at end-diastole, as measured using B-mode echocardiography. LV thickness was calculated from the short-axis view, and the mean values of the thickest segment obtained in three consecutive cardiac cycles were used [28].

Furthermore, 229 additional cats (95 Munchkin, 132 Scottish Fold, and 2 Minuet) without a known phenotypic myocardium status were used for allele frequency analysis. The cats were either client-owned with genetic testing done at Anicom Pafe Inc. (Japan) or were neutered at Shinjuku Gyoenmae Animal Hospital. These additional cats were not associated with any specific disease and were included as the general group. All owners provided informed consent to use their cat-specific data for scientific research.

## Genotyping

The genomic DNA of each cat was extracted from whole blood or buccal swab samples or reproductive tissues removed by castration. DNeasy Blood & Tissue Kit (Qiagen, Netherlands) was used for DNA extraction from reproductive tissues and blood according to the manufacturer's instructions. Likewise, Chemagic ™ DNA Buccal Swab Kit (PerkinElmer, U.S.) and DNAdvance Kit (Beckman Coulter, U.S.) were used for DNA extraction from the oral mucosal tissue. The extracted DNA samples were used to identify *MYBPC3* (p.A31P, p.R820W, and p. A74T), *MYH7* (p.E1883K), and *ALMS1* (p.G3376R) variants. The genotypes of *MYBPC3* p. A31P and p.R820W were confirmed by the Taqman assay, while the others were identified by Sanger sequencing (primers and probes are shown in S1 Table) performed at Eurofins Genomics Inc. (Tokyo, Japan). The obtained DNA sequences were aligned using MEGA 7: Molecular Evolutionary Genetics Analysis version 7.0 for bigger datasets [29]. The genotype of each sample was classified as wild-type (WT), heterozygous, or homozygous.

## *ALMS1* variant detection using public data

The publicly available whole genome data (as fastq files) were downloaded from Sequence Read Archive (SRA, https://www.ncbi.nlm.nih.gov/sra). The downloaded data included those of 40 cats, which comprised 30 purebred cats belonging to 13 breeds and 10 random bred cats. Quality filtering of the raw fastq files was performed using trim_galore v 0.6.5 with default settings. The DRAGEN software v. 3.6 (Illumina Inc.) was used for mapping to the domestic cat genome (felCat9 [30]) and variant calling. The *ALMS1* variant (A3:92,439,157) was extracted using Vcftools v 0.1.16 [31].

## Statistical analysis

Differences between the HCM-affected group and the general group in terms of the presence of the *ALMS1* variant, within the same breed, were detected using Fisher's exact test using R v 4.1.2 [32]. χ-square test was not applied due to the small number of cells sampled in the 2 × 3 contingency table.

## Results

### Detection of genetic variants in echocardiographically examined cats

The five genetic variants were genotyped in cats that underwent echocardiography, detecting one heterozygous cat for *MYBPC3* p.A31P variant and nine heterozygous cats for *ALMS1* p. G3376R variant among 57 HCM-affected cats (Table 1). *MYBPC3* p.A31P was identified in one of the four Munchkin cats, whereas it was not found in six Maine Coon cats. The *ALMS1* p.G3376R variant was detected in three breeds: Scottish Fold (7/18 cats), Exotic Shorthair (1/3 cats), and Sphynx (1/1 cat). The HCM-affected Scottish Fold accounted for the highest proportion of HCM-affected cats in this study, which ranged from 7 months to 14 years of age (mean of 5.4 years±4.6 SD). Among these, the *ALMS1* variant carriers included a wide range of cats aged between 8 and 14 years (mean of 7 years±5.1 SD); and there was no association between the *ALMS1* variant and age. Additionally, a six-month-old Scottish Fold (1/1) cat was heterozygous for the *MYBPC3* p.A31P variant, and two American Shorthair (2/3) cats, aged 1 and 3, were heterozygous for the *ALMS1* p.G3376R variant, among the 19 non-HCM cats (Table 2).

These results indicate the presence of the variants *MYBPC3* p.A31P and *ALMS1* p. G3376R in two and three new cat breeds, respectively. In addition, no homozygous *MYBPC3* or *ALMS1* variants were detected in the cats examined by echocardiography. Moreover, *MYBPC3* p.R820W, p.A74T, and *MYH7* p.E1883K were not included (Tables 1 and 2). The phenotypes obtained using echocardiography and the genotype of each variant in individual cats are presented in S2 Table. The nine *ALMS1* p.G3376R positive cats were classified with seven cats in stage B1, with one in stage B2, and with one in stage C according to the American College of Veterinary Internal Medicine (ACVIM) stage classification [9]. The stage C cat showed with pleural effusion and pulmonary edema, and had syncope after excitement. Other phenotypes included four of the *ALMS1* p.G3376R positive cats with left ventricular outflow tract obstruction. One *MYBPC3* p.A31p positive cat was classified as stage B1 and had no left ventricular outflow tract obstruction. Based on the observed phenotypic variations among cats, it proved challenging to ascertain the precise association of each mutation with a particular stage of HCM.

### Detection of genetic variants in non-echocardiographically examined cats

The two variants (*MYBPC3* p.A31P and *ALMS1* p.G3376R) detected in the first genotypic analysis were used during the second genotypic analysis of non-echocardiographically examined cats. The *MYBPC3* variant was identified in Munchkin and Scottish Fold cats, while the HCM-affected Scottish Fold cats exhibited a high prevalence of the *ALMS1* variant. Therefore, approximately 100 cats from each of these breeds (Munchkin; N = 95, and Scottish Fold; N = 132) were randomly selected from the general group unrelated to the specific disease and examined for the prevalence of each variant. As a result, two Munchkin and four Scottish Fold heterozygous cats for *MYBPC3* p.A31P variant were detected (Table 3). The mutated allele frequencies of Munchkin and Scottish Fold breeds in the general group were 1.05% and 1.51%, respectively. Similarly, one homozygous and 22 heterozygous Scottish Fold cats carrying *ALMS1* p.G3376R variant were detected in the general group (Table 3), and the allelic

**Table 1. The number of HCM-associated variant carriers in the HCM-affected group.**

| Breed | N | MYBPC3 p.A31P | MYBPC3 p.R820W | MYBPC3 p.A74T | MYH7 p.E1883K | ALMS1 p.G3376R |
|-------|---|------|------|------|------|------|
| | | | | Total positive | | |
| American Shorthair | 1 | 0 | 0 | 0 | 0 | 0 |
| Bengal | 4 | 0 | 0 | 0 | 0 | 0 |
| British Shorthair | 5 | 0 | 0 | 0 | 0 | 0 |
| Exotic Shorthair | 3 | 0 | 0 | 0 | 0 | 1[*] |
| Maine Coon | 6 | 0 | 0 | 0 | 0 | 0 |
| Munchkin | 4 | 1[*] | 0 | 0 | 0 | 0 |
| Norwegian Forest Cat | 5 | 0 | 0 | 0 | 0 | 0 |
| Ragdoll | 2 | 0 | 0 | 0 | 0 | 0 |
| Random bred | 5 | 0 | 0 | 0 | 0 | 0 |
| Russian Blue | 1 | 0 | 0 | 0 | 0 | 0 |
| Scottish Fold | 18 | 0 | 0 | 0 | 0 | 7[*] |
| Siberian Forest Cat | 1 | 0 | 0 | 0 | 0 | 0 |
| Singapura | 1 | 0 | 0 | 0 | 0 | 0 |
| Sphinx | 1 | 0 | 0 | 0 | 0 | 1 |
| Total | 57 | 1 | 0 | 0 | 0 | 9 |

* indicates that the variant has never been found before in the breed.

frequency of the variant was 9.09%. In addition, the prevalence of the ALMS1 variant in Scottish Fold cats was compared between the HCM-affected and general groups, and a statistical test was performed (Fig 1). It was proposed that a strong effect of the ALMS1 variant on HCM development would indicate a significantly higher prevalence of this variant in the HCM-affected group than in the general group, although the difference between the two groups (Fisher's exact test; $P = 0.083$) was insignificant. Next, considering the possibility that the ALMS1 variant was not fully identified during the first analysis because of the small population size, we genotyped the variant in Munchkin cats, resulting in the detection of 16 Munchkin cats heterozygous for ALMS1 p.G3376R in the general group (Table 3), and the mutated allele frequency was 8.42%. Furthermore, an ALMS1 variant heterozygote was identified in the Minuet cat, a breed closely related to Munchkin (Table 3). Together, these results indicate the presence of ALMS1 p.G3376R variant in five new breeds other than Sphinx.

**Table 2. The number of HCM-associated variant carriers in the non-HCM group.**

| Breed | N | MYBPC3 p.A31P | MYBPC3 p.R820W | MYBPC3 p.A74T | MYH7 p.E1883K | ALMS1 p.G3376R |
|-------|---|------|------|------|------|------|
| | | | | Total positive | | |
| American Shorthair | 3 | 0 | 0 | 0 | 0 | 2[*] |
| Maine Coon | 6 | 0 | 0 | 0 | 0 | 0 |
| Munchkin | 1 | 0 | 0 | 0 | 0 | 0 |
| Random bred | 8 | 0 | 0 | 0 | 0 | 0 |
| Scottish Fold | 1 | 1[*] | 0 | 0 | 0 | 0 |
| Total | 19 | 1 | 0 | 0 | 0 | 2 |

* indicates that the variant has never been found before in the breed.

**Table 3. The number of HCM-associated variant carriers among the Munchkin, Scottish Fold, and Minuet cats in the general group.**

| | | Total positive | |
|---|---|---|---|
| **Breed** | **N** | *MYBPC3* **p.A31P** | *ALMS1* **p.G3376R** |
| Munchkin | 95 | 2 | 16 |
| Scottish Fold | 132 | 4 | 23[*] |
| Minuet | 2 | 0 | 1[*] |

The one ALMS1 p.G3376R positive Scottish Fold was a homozygous cat, and the others are heterozygous cats. Other positive cats were heterozygous.

[*] indicates that the variant has never been found before in the breed.

Finally, we identified the *ALMS1* genotype from the publicly available cat genome to ascertain the global geographic distribution of *ALMS1* p.G3376R. This dataset included purebred cats and random bred cats (Domestic Shorthair) of American, European, Asian, and Middle Eastern origin. Two heterozygous cats (one American Shorthair cat and one random bred cat) were detected, and both were of US origin (S3 Table).

The *ALMS1* variant genotypes of each cat among the 18 HCM-affected and the 132 general Scottish Fold cats, were evaluated. White, gray, and black bars indicate wild-type, heterozygous, and homozygous variants, respectively. There is no significant difference between the proportion of the two groups (Fisher's exact test; P = 0.083). The detailed number of cats are presented in Tables 1 and 3.

## Discussion

This study focused on determining the widespread occurrence of genetic variants associated with feline HCM in cat breeds, using HCM-affected, non-HCM, and general cat populations in Japan. *MYBPC3* p.A31P was present in at least two non-Maine Coon breeds, and *ALMS1* p.G3376R was detected in more than five non-Sphynx breeds. Therefore, the genetic variants involved in HCM development may not be limited to specific breeds, as previously reported.

### Expansion of breeds with *MYBPC3* p.A31P mutation

HCM-associated mutations in cats are usually considered breed-specific variants [10, 17, 18, 33]. Several previous studies have reported the specific association of *MYBPC3* p.A31P variant with the Maine Coon breed; this variant is absent in other breeds including interbreed cats [20, 25, 33]. As an exception, carriers of this variant were found in British Longhair, but only in one of the two cats [26]. This heterozygous cat is the only *MYBPC3* p.A31P positive non-Maine Coon cat for which data was available. Interestingly, our study is the first study to show the presence of this HCM-associated variant in three Munchkin and five Scottish Fold cats, expanding the range of breeds carrying the *MYBPC3* p.A31P mutation. However, *MYBPC3* p.A31P variant could not be the primary cause of HCM in these cats because of its low penetrance in HCM-affected breeds [34]. Additionally, one carrier Scottish Fold cat was not diagnosed with HCM, possibly because of its young age of five months at the time of diagnosis. The cat may develop HCM in the future since the preferred age for manifestation of HCM symptoms is about two to three years [8]. Moreover, several cats heterozygous for this variant have been reported to be HCM-unaffected [19]. A meta-analysis of numerous Maine Coon cats suggested an increased prevalence risk only in the homozygous condition [18]. These studies further support the presence of *MYBPC3* p.A31P heterozygous cats in the non-HCM group. This study could not provide evidence for an association between the *MYBPC3* p.A31P

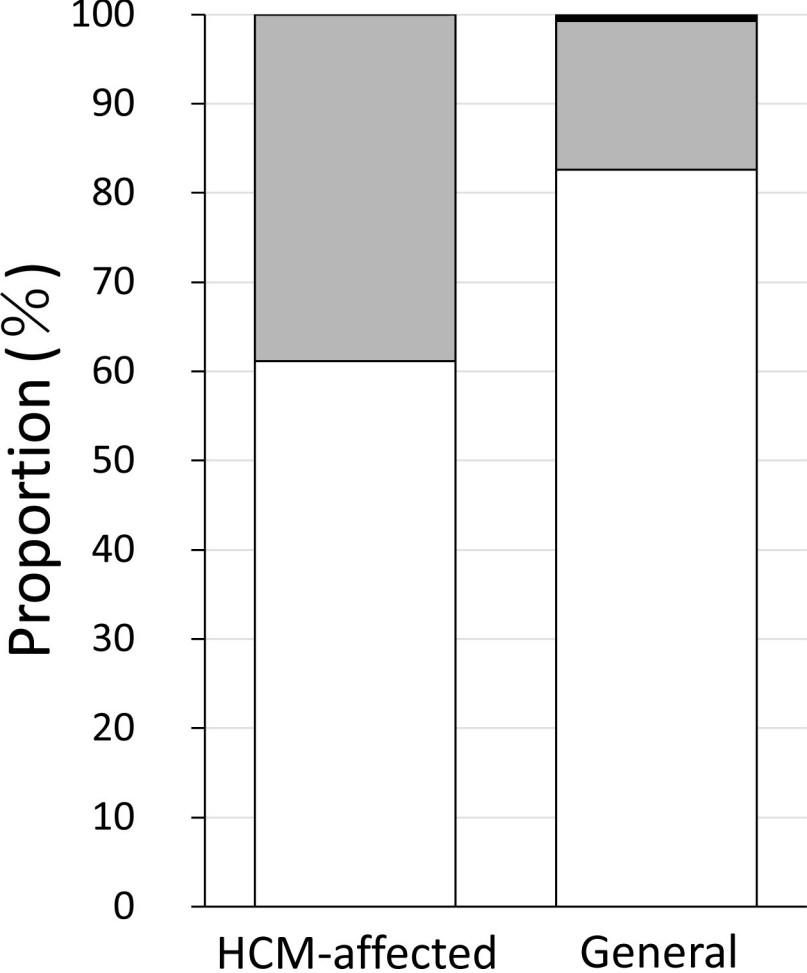

**Fig 1. Comparison of the proportion of *ALMS1* variant carriers in the Scottish Fold cats.**

variant and the onset of HCM in non-Maine Coon breeds. However, it is clinically significant that a new non-Maine Coon breed with the *MYBPC3* p.A31P variant, an established causative mutation for feline HCM, has been identified.

## Association between *ALMS1* p.G3376R and HCM onset

*ALMS1* p.G3376R is a Sphynx-specific mutation that has been reported in a few carriers of over 200 non-Sphynx cats without known heart disease [23]. To the best of our knowledge, no other studies have associated *ALMS1* with HCM in cats, and *ALMS1* has not been shown to be as clearly associated with HCM as *MYBPC3*. This study reports the presence of *ALMS1* variant in more diverse cat breeds than that reported previously. The allelic frequency of this mutation in Munchkin and Scottish Fold cats is moderate, suggesting that this variant is not specific to the Sphynx breed. In addition, on comparing the proportion of carriers of this variant in the HCM-affected and general-group Scottish Fold cats, no association between the variant and HCM development was observed. Previous studies have shown a high prevalence of this variant in the HCM-affected Sphynx breed (87.3%), although Sphynx cats from the HCM-unaffected and general population were not used [23]. Therefore, the possibility of widespread distribution of this variant in Sphynx cats unrelated to HCM cannot be excluded. However,

the effect of *ALMS1* variant can probably be diluted under the influence of a genetic background different from that of the Sphynx. Furthermore, only 1 homozygote and 38 heterozygote cats carrying the *ALMS1* variant were observed among the 227 cats in the general group. The ratio of homozygotes was very low compared to that in a previous study in which 35 homozygotes and 27 heterozygotes were detected among 62 HCM-affected Sphynx cats [23]. Therefore, the homozygous *ALMS1* p.G3376R variant could influence HCM development. Considering these findings, the association of *ALMS1* variant with HCM disease should be carefully evaluated in more cat breeds.

## The genetic history of *MYBPC3* and *ALMS1* variants

In this study, we identified HCM-related variants in breeds not previously reported. Since *MYBPC3* p.A31P has not been detected in many breeds [20, 25, 33], its phylogenetic transmission from Maine Coon to a relatively new breed, such as Munchkin, over a long period is unlikely. In addition, the possibility of a *de novo* mutation in the same nucleotide in each breed cannot be excluded; however, it is minimal. Previous studies have shown that genomes of some Japanese Munchkin and Scottish Fold cats share the genetic component with Maine Coon breed [27]. These findings suggest outcrossing of other breed cats with *MYBPC3* p. A31P-carrying Maine Coon cats. A long-hair trait further indicates the outcrossing between breeds. This is consistent with a previous study showing the presence of *MYBPC3* p.A31P variant in British Longhair cats, suggesting the occurrence of Maine Coon breed in British Longhair historical lineage due to the similarity with the longhair breed [26]. Unfortunately, the information regarding cat hair length was unavailable in our study. Nevertheless, Munchkin and Scottish Fold contain long-haired cats; therefore, these cats carrying the *MYBPC3* variant may share similar features. Other pure-bred cats in the Japanese population with a genetic structure similar to the Maine Coon cat include the American Curl, Persian, Siberian Forest Cat, and Norwegian Forest Cat [27]. Additional genetic analysis using a higher number of cats in these breeds may help identify *MYBPC3* variant carriers and genetic history of the variants. In contrast, the theory behind the widespread occurrence of *ALMS1* p.G3376R in several breeds is unclear. Considering a large gap in the appearance phenotype between the breeds, including the genetic variant-carrying Sphinx, it excludes the possibility of hybridization in recent history. The breeds that carried the *ALMS1* variant in this study are of European or American origin [35, 36]. The *ALMS1* variant, examined using publicly available cat genomes based on next-generation sequencing, was not detected in any Asian or Middle Eastern cat breeds (S3 Table). Therefore, a common ancestor of Western origin may have initially possessed the causative factor. Interestingly, the *ALMS1* variant was not identified in the Scottish Fold cat in a previous study [23], suggesting the specific prevalence of the Scottish Fold breed carrying the *ALMS1* variant in the Japanese population. Several Scottish Fold lines have different genetic structures between the Japanese and U.S. populations [27], which could be the cause of the difference in allelic frequencies of the *ALMS1* variant.

## Importance of investigating pathogenesis across a wide range of cat breeds and populations

This study highlights the importance of investigating the cross-breed distribution of heritable pathogenesis across cat populations. The feline HCM guidelines do not recommend genetic testing for *MYBPC3* p.A31P variant in non-Maine Coon breeds because of the breed-specificity of this variant [9]. However, the variant was identified in non-Maine Coon breeds in this study (e.g., Munchkin and Scottish Fold), indicating a possibility for variant-mediated onset of HCM. Therefore, genetic testing of *MYBPC3* p.A31P in non-Maine Coon breeds is a valid

measure for HCM diagnosis and to prevent the establishment of HCM-affected colonies. Besides, the variants *MYBPC3* p.A74T, p.R820W, and *MYH7* p.E1883K were not detected in any cat in this study. The two variants *MYBPC3* p.R820W and *MYH7* p.E1883K have been reported only in Ragdoll cats and a Domestic Shorthair, respectively [20, 33]. However, a number of studies in the United States and Europe have identified *MYBPC3* p.A74T in several breeds [19, 20, 37], supporting the importance of investigating genetic pathogenesis in different populations. This study has certain limitations, such as the younger age of the animals and the relatively small sample sizes; therefore, future research with a larger sample size would enable identifying a wider distribution of these three variants in cats. Meanwhile, a species-exhaustive study of known variants would not be sufficient to understand HCM development comprehensively. Notably, 47 out of 57 HCM-affected cats in this study did not carry any of the five variants associated with HCM. In humans, over 450 causative variants have been identified in the genes associated with sarcomeres and myofilaments related to the genetic disease HCM [38–40]. Hence, a variety of HCM-associated variants may still be identified in cats, and the prevalence of such variants should be analyzed across breeds and populations.

## Conclusion

This study demonstrated that variants associated with feline HCM are found in various non-specific cat breeds. The *MYBPC3* p.A31P variant could be responsible for the onset of HCM in Scottish Fold and Munchkin breeds, and the *ALMS1* p.G3376R variant may not occurred in a Sphynx-specific manner. The study highlights the significance of examining the prevalence of genetic variants across different populations and breeds.

## Supporting information

**S1 Table. Primers and Taqman probes list.** The sequences of used primers and Taqman probes were shown.
(XLSX)

**S2 Table. Diagnosis information and genotype of each cat.** *: The stage classification method was from previous paper [9].
(XLSX)

**S3 Table. *ALMS1* variant detection from public data.** Origins of Domestic Shorthair according to each published data. SRA Run ID denote Run ID of Sequence Read Archive (https://www.ncbi.nlm.nih.gov/sra).
(XLSX)

## Acknowledgments

We would like to thank all Japanese breeders, veterinarians, and cat owners who participated in this study for collecting tissues and buccal swabs from their cats. We also thank all the staff performing genetic testing at the Anicom Specialty Medical Institute Inc. for their support. We would like to thank Editage (www.editage.com) for English language editing.

## Author Contributions

**Conceptualization:** Ryohei Suzuki, Yuki Matsumoto.

**Data curation:** Noriyoshi Akiyama, Ryohei Suzuki, Hisashi Ukawa.

**Formal analysis:** Noriyoshi Akiyama, Takahiro Saito, Yuki Matsumoto.

**Funding acquisition:** Yuki Matsumoto.

**Investigation:** Noriyoshi Akiyama, Ryohei Suzuki, Takahiro Saito, Yunosuke Yuchi, Hisashi Ukawa, Yuki Matsumoto.

**Project administration:** Ryohei Suzuki, Yuki Matsumoto.

**Resources:** Ryohei Suzuki, Takahiro Saito, Yunosuke Yuchi, Hisashi Ukawa, Yuki Matsumoto.

**Supervision:** Ryohei Suzuki, Yuki Matsumoto.

**Visualization:** Noriyoshi Akiyama.

**Writing – original draft:** Noriyoshi Akiyama, Ryohei Suzuki, Takahiro Saito.

**Writing – review & editing:** Noriyoshi Akiyama, Ryohei Suzuki, Yunosuke Yuchi, Yuki Matsumoto.

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
