## [Decision Letter · Decision Letter 0]

9 Feb 2023

PONE-D-23-01537Presence of Known Feline ALMS1 and MYBPC3 Variants in a Diverse Cohort of Cats with Hypertrophic Cardiomyopathy in JapanPLOS ONE

Dear Dr. Matsumoto,

Thank you for submitting your manuscript to PLOS ONE. After careful consideration, we feel that it has merit but does not fully meet PLOS ONE’s publication criteria as it currently stands. Therefore, we invite you to submit a revised version of the manuscript that addresses the points raised during the review process.

We look forward to receiving your revised manuscript.

Kind regards,

Shinsuke Yuasa

Academic Editor

PLOS ONE

Journal Requirements:

"This study was supported by Research Foundation of Anicom Specialty Medical Institute Inc. (Japan), Anicom Pafe Inc. (Japan) and Nippon Veterinary and Life Science University."

3. We noted in your submission details that a portion of your manuscript may have been presented or published elsewhere. 

"We have submitted our manuscript to the "Animals Genetics", however, the editor has suggested that it is not suitable for publication as an Original article. Despite this, they have recommended its publication as a Short communication. Nonetheless, we deem that the discussion presented in this manuscript is of significant importance for Original Article publication, thus we have decided to submit it to PLOS ONE."

Reviewers' comments:

Reviewer's Responses to Questions

**Comments to the Author**

1. Is the manuscript technically sound, and do the data support the conclusions?

Reviewer #1: Partly

Reviewer #2: Yes

2. Has the statistical analysis been performed appropriately and rigorously? 

Reviewer #1: No

Reviewer #2: Yes

3. Have the authors made all data underlying the findings in their manuscript fully available?

Reviewer #1: Yes

Reviewer #2: No

4. Is the manuscript presented in an intelligible fashion and written in standard English?

Reviewer #1: No

Reviewer #2: Yes

5. Review Comments to the Author

Reviewer #1: SUMMARY:

The authors investigated the clinical prevalence of HCM gene variants in cats. They reported rare occurrences of MYBPC3 p.A31P and ALMS1 p.G3376R in specific cat breeds which have not been reported yet. Although this report has important implications for the future application of genetic testing in cats, several points must be addressed.

GENERAL COMMENTS:

The contents of the manuscript should be stated in an appropriate order. For example, limitations and conclusions are not listed where they should be. That makes the manuscript confusing. Also, the reviewer highly recommends the authors make subtitles for each subsection in the Discussion, and the first subsection should be the summary of findings in the present study. These modifications would make the manuscript easier to comprehend.

Whereas MYBPC3 p.A31P is an established genetic mutation in HCM, ALMS1 p.G3376R is not established (the association with HCM is just suggested). These two findings should not be listed in the same importance. The authors should state the clinical implication of these findings in each subsection, or in each paper.

The last paragraph of the Introduction contains the results of this investigation. The author should move these descriptions to the discussion section, or conclusions section (create it after the discussion section).

In the last sentence of page 16, the expression “genetic testing of MYBPC3 p.A31P in non-Maine Coon breeds is necessary for HCM diagnosis and to prevent the establishment of HCM-affected colonies”, seems too strong for the suggestion from results with a relatively small number of subjects. The author would be advised to alter the expression in an appropriate manner.

MINOR COMMENTS:

(Discussion, Page 14, Line 13) The author should add “Bell et al., 2012” into the Reference.

(Table 1) As per the manuscript, the MYBPC3 p.A31P in Munchkin is the variant that has never been found. Therefore, an asterisk should be attached next to the number “1”.

(Table 3) There is no row about Minuet. As per the manuscript, one heterozygote ALMS1 variant was found in the 2 Minuet cats. If the author considers the ALMS1 variant in Minuet is worth reporting, it should be listed also in the Table.

Figure 1 seems not to have enough information. It should be transformed into another table, which enables the authors to fill out the absolute numbers of each zygosity in each group. They can also attach percentages next to each absolute number.

Reviewer #2: The authors of the article present the results of genetic study of five variants previously associated with HCM in 57 HCM-affected, 19 HCM-unaffected, and 227 non-examined cats from the Japanese population to investigate the occurrence among breeds. They conclude that two of these variants could not be limited to specific breeds, as previously it has been reported.

- Abstract

You should check the variant name p.R830W.

- Introduction

At the end of this section you describe your results and conclusion. You shouldn’t include your results in the introduction Section.

- Methods

The authors include 2 Minuet cats without a cardiac phenotype in “Phenotyping and selected animals” section, nevertheless the genetic results are not provided. The authors should include the genetics results of these cats in table 3.

In this section the authors must detail the statistical analysis used.

- Results

I suggest to the authors that they include a table with the main clinical variables of the cats studied such as breed, age, age of phenotype, gender, mm hypertrophy, events or arrhythmia.

The genetic results could be integrated and summarized in a table that includes the frequency of the variants and the comparison between the different groups of animals.

6. PLOS authors have the option to publish the peer review history of their article (what does this mean?). If published, this will include your full peer review and any attached files.

Reviewer #1: **Yes: **Keitaro Akita

Reviewer #2: No

---

## [Author Response · Author response to Decision Letter 0]

20 Feb 2023

We wish to re-submit the manuscript titled, “Presence of known feline ALMS1 and MYBPC3 variants in a diverse cohort of cats with hypertrophic cardiomyopathy in Japan.” The manuscript ID is PONE-D-23-01537.

We thank you and the reviewers for your thoughtful suggestions and insights. The manuscript has benefited from these insightful suggestions. I look forward to working with you and the reviewers to move this manuscript closer to publication in the PLOS ONE.

The manuscript has been rechecked and the necessary changes have been made in accordance with the reviewers’ suggestions. The responses to all comments have been prepared and attached herewith/given below. The changes made in the manuscript are tracked using the track changes option in MS word; we have submitted two versions of the revised manuscript, one with changes tracked and the other without any markup, as instructed in the decision letter.

---

## [Decision Letter · Decision Letter 1]

1 Mar 2023

PONE-D-23-01537R1Presence of Known Feline ALMS1 and MYBPC3 Variants in a Diverse Cohort of Cats with Hypertrophic Cardiomyopathy in JapanPLOS ONE

Dear Dr. Matsumoto,

Thank you for submitting your manuscript to PLOS ONE. After careful consideration, we feel that it has merit but does not fully meet PLOS ONE’s publication criteria as it currently stands. Therefore, we invite you to submit a revised version of the manuscript that addresses the points raised during the review process.

We look forward to receiving your revised manuscript.

Kind regards,

Shinsuke Yuasa

Academic Editor

PLOS ONE

Journal Requirements:

Reviewers' comments:

Reviewer's Responses to Questions

**Comments to the Author**

1. If the authors have adequately addressed your comments raised in a previous round of review and you feel that this manuscript is now acceptable for publication, you may indicate that here to bypass the “Comments to the Author” section, enter your conflict of interest statement in the “Confidential to Editor” section, and submit your "Accept" recommendation.

Reviewer #1: All comments have been addressed

2. Is the manuscript technically sound, and do the data support the conclusions?

Reviewer #1: Partly

3. Has the statistical analysis been performed appropriately and rigorously? 

Reviewer #1: Yes

4. Have the authors made all data underlying the findings in their manuscript fully available?

Reviewer #1: Yes

5. Is the manuscript presented in an intelligible fashion and written in standard English?

Reviewer #1: Yes

6. Review Comments to the Author

Reviewer #1: The authors have partially addressed the questions and comments raised in the last revision. There are a few points that must be addressed.

The aim of this study was removed from the whole manuscript. It should be retained in the last part of the introduction.

The reviewer still recommends that the authors should state the conclusion at the end of the manuscript. It should the main message to the reader clearer.

All the abbreviations in the supplemental tables should be spelled out in the footnotes.

Although the authors added the clinical information of subjects in the supplemental table 2, they did not describe it in either the results or the discussion in detail. They should describe the clinical characteristics or cardiac events specific to MYBPC3 p.A31p positive cases or ALMS1p.G3376R positive cases.

7. PLOS authors have the option to publish the peer review history of their article (what does this mean?). If published, this will include your full peer review and any attached files.

Reviewer #1: **Yes: **Keitaro Akita

---

## [Author Response · Author response to Decision Letter 1]

5 Mar 2023

From Reviewer 1

> The aim of this study was removed from the whole manuscript. It should be retained in the last part of the introduction.

Response: Thank you for your suggestion. We retained the aim of our study in the last part of the introduction (Lines 80 to 82).

> The reviewer still recommends that the authors should state the conclusion at the end of the manuscript. It should the main message to the reader clearer.

Response: We agree with your suggestion. We added conclusion section in Lines 313 to 318.

> All the abbreviations in the supplemental tables should be spelled out in the footnotes.

Response: We followed your advice and added column name in S2 table (“y” and “m” in age column, and ACVIM in “Heart failure stage” column), and footnote to S3 table (SRA Run ID for Line 439-440).

＞Although the authors added the clinical information of subjects in the supplemental table 2, they did not describe it in either the results or the discussion in detail. They should describe the clinical characteristics or cardiac events specific to MYBPC3 p.A31p positive cases or ALMS1p.G3376R positive cases.

Response: Thank you for your suggestion. We added description regarding clinical information in results section (Lines 163 to 172).

---

## [Decision Letter · Decision Letter 2]

10 Mar 2023

Presence of known feline ALMS1 and MYBPC3 variants in a diverse cohort of cats with hypertrophic cardiomyopathy in Japan

PONE-D-23-01537R2

Dear Dr. Matsumoto,

We’re pleased to inform you that your manuscript has been judged scientifically suitable for publication and will be formally accepted for publication once it meets all outstanding technical requirements.

Kind regards,

Shinsuke Yuasa

Academic Editor

PLOS ONE

Additional Editor Comments (optional):

Reviewers' comments:

Reviewer's Responses to Questions

**Comments to the Author**

1. If the authors have adequately addressed your comments raised in a previous round of review and you feel that this manuscript is now acceptable for publication, you may indicate that here to bypass the “Comments to the Author” section, enter your conflict of interest statement in the “Confidential to Editor” section, and submit your "Accept" recommendation.

Reviewer #1: All comments have been addressed

2. Is the manuscript technically sound, and do the data support the conclusions?

Reviewer #1: Yes

3. Has the statistical analysis been performed appropriately and rigorously? 

Reviewer #1: Yes

4. Have the authors made all data underlying the findings in their manuscript fully available?

Reviewer #1: Yes

5. Is the manuscript presented in an intelligible fashion and written in standard English?

Reviewer #1: Yes

6. Review Comments to the Author

Reviewer #1: The authors have successfully addressed the comments raised in the last revision. The reviewer has no further comments.

7. PLOS authors have the option to publish the peer review history of their article (what does this mean?). If published, this will include your full peer review and any attached files.

Reviewer #1: **Yes: **Keitaro Akita

---

## [Editor Report · Acceptance letter]

10 Apr 2023

PONE-D-23-01537R2 

Presence of known feline *ALMS1* and *MYBPC3* variants in a diverse cohort of cats with hypertrophic cardiomyopathy in Japan 

Dear Dr. Matsumoto:

I'm pleased to inform you that your manuscript has been deemed suitable for publication in PLOS ONE. Congratulations! Your manuscript is now with our production department. 

Kind regards, 

on behalf of

Dr. Shinsuke Yuasa 

Academic Editor

PLOS ONE